# Arbuscular Mycorrhizal Fungi Promote Physiological and Biochemical Advantages in *Handroanthus serratifolius* Seedlings Submitted to Different Water Deficits

**DOI:** 10.3390/plants11202731

**Published:** 2022-10-15

**Authors:** Tatiane Santos Correia, Túlio Silva Lara, Jéssica Aires dos Santos, Ludyanne da Silva Sousa, Marcos Diones Ferreira Santana

**Affiliations:** Institute of Water Science and Technology, Federal University of Western Para, Santarém 68040-255, Brazil

**Keywords:** mycorrhizal association, plant physiology, AM fungi, osmoregulation, proline, yellow ipê

## Abstract

Climate change causes increasingly longer periods of drought, often causing the death of plants, especially when they are in the early stages of development. Studying the benefits provided by arbuscular mycorrhizal (AM) fungi to plants in different water regimes is an efficient and sustainable strategy to face climate change. Thus, this study investigated the influence of AM fungi on *Handroanthus serratifolius* seedlings under different water regimes, based on biochemical, and nutritional growth parameters. The experiment was carried out in *H. serratifolius* seedlings cultivated with mycorrhizas (+AMF) and without mycorrhizas (-AMF) in three water regimes; a severe water deficit (SD), a moderate water deficit (MD), and a well-watered (WW) condition. AM fungi provided greater osmoregulation under water deficit conditions through the accumulation of soluble sugars, total free amino acids, and proline, as well as by reducing sugar. The increase in the absorption of phosphorus and nitrate was observed only in the presence of fungi in the well-watered regimen. A higher percentage of colonization was found in plants submitted to the well-watered regimen. Ultimately, AM fungi promoted biochemical, nutritional, and growth benefits for *H. serratifolius* seedlings under the water deficit and well-hydrated conditions, proving that AMF can be used to increase the tolerance of *H. serratifolius* plants, and help them to survive climate change.

## 1. Introduction

In recent decades, with the increase in global warming and climate change, water scarcity has increased considerably [1] and has become one of the main problems in world agriculture, as it directly affects crop development and yield [2]. However, the negative effects of water scarcity also affect tropical forests, causing high tree mortality [3], in addition to strongly altering the regional carbon balance, thereby accelerating and making the effects of negative climate change even more dramatic [4]. Forest replacement or recovery efforts can help to conserve regions affected by deforestation, for example, and contribute to the minimization of these effects, but plants in the growth phase are even more susceptible to water deficit than adult plants [5,6,7].

Water deficits stimulate the accumulation of abscisic acid, which is responsible for closing the stomata, reducing gas exchange and consequently limiting photosynthesis, which can cause death by carbon starvation and total depletion of plant reserves [8]. It also causes a drop in the water potential of the plant, which can result in xylem cavitation; this occurs when liquid water passes into the vapor phase within the xylem, resulting in the formation of air bubbles that then cause embolisms [9,10].

A good solution for this issue is to explore the influence of arbuscular mycorrhizal (AM) fungi on the development of forest species under well-watered and water stress conditions [11,12]. These fungi improve the plant’s growth and development [13] due to greater absorption of nutrients, mainly phosphorus (P) and nitrogen (N) [14,15], and by encouraging biochemical changes, such as osmotic adjustment from the accumulation of organic and inorganic solutes, such as proline, soluble sugars, total free amino acids, and ions [16]. Accumulation of these solutes is a strategy plants use to better tolerate water stress, as the accumulation reduces cellular osmotic potential, and increases cellular water retention during water stress [17], keeping the stomata open, and, consequently retaining gas exchange. The solutes act as osmoprotective agents.

Among the forest species suitable for environmental recovery, *Handroanthus serratifolius* (Vahl) SO Grose, popularly known as yellow ipê, stands out. This species is native to South America, with a wide distribution in Brazil and a long history of use in reforestation and landscaping. The wood has high commercial value, while the leaves and flowers present pharmacological potential [18,19,20,21]. Despite its importance, there are very few reports in the literature of studies involving AM fungi and *H. serratifolius*. There are some studies with the genus *Handroanthus* demonstrating that AMF inoculation provided an increase in height, biomass, and calcium and nitrogen content by up to 10%. [22,23].

However, the choice of AM fungi may be a determining factor in the interaction between AMF-plant species. As a result, the savanna area, located in the west of Pará state, Brazil, surrounded by the Amazon Forest, an area characterized by a period of drought and nutrient-poor soils, was chosen for the bioprospecting of native AM fungi. In addition, there is a lack of studies aimed at understanding the symbiotic relationship of the AM fungi with *H. serratifoliu* species.

Thus, the aims of this work are; (i) to study the influence of AM fungi on the accumulation of osmolytes derived from nitrogen and carbohydrate metabolism in *H. serratifolius* under different water regimes; (ii) to evaluate the influence of AM fungi on the absorption of phosphorus and nitrogen compounds in *H. serratifolius* under different water regimes; (iii) to evaluate the influence of AMF on the growth and development of *H. serratifolius* under different water regimes. Therefore, we hypothesized that inoculation with AM fungi will have a beneficial impact on the tolerance of *H. serratifolius* seedlings to water deficits by promoting a greater accumulation of total free amino acids, proline, and some soluble carbohydrates.

## 2. Results

### 2.1. Growth and Colonization

In general, the +AMF plants showed a greater increase in shoot dry mass than the -AMF plants, around 18% in WW, 14% in MD, and 4% in SD. An interesting result was observed in plants under +MD that showed an increase of 17% in shoot dry mass in relation to plants under -WW (Table 1). The height of yellow ipê plants was more strongly influenced by AMF in WW, where +AMF plants provided an increase of about 11%. Inoculation with AMF and the different water regimes did not provide significant differences (*p* ≤ 0.05) in leaf area and total leaf area (Table 1). The root volume was positively influenced by the water regime, where plants under WW showed a volume increase of about 50% in relation to plants under MD and SD. However, the presence of AMF did not significantly influence the water regimes (Table 1). The under -AMF plants showed no colonization. Plants inoculated with AMF showed Arum-type colonization; the percentage of colonization (%C) was directly related to the water regime, where the highest %C was observed in plants under +WW, and the lowest was observed in plants under +SD (Table 1). As observed in the dry mass, the height of plants in +MD was also 12% higher than in -WW plants (Table 1).

### 2.2. Nitrogen Metabolism

Plants in -SD and -MD showed accumulations of total proteins at about 48% and 26%, higher than in +SD and +MD treatments, respectively. In WW, there was no statistical difference (*p* ≤ 0.05) (Figure 1A). However, +AMF plants showed greater accumulation of total amino acids compared to -AMF plants regardless of the water regime (Figure 1B). The plants in +SD and +MD showed a higher accumulation than the plants in -SD and -MD, about 20% and 40%, respectively, while in conditions of good hydration the +WW plants showed an accumulation that was 15% higher than the -WW plants.

The +AMF plants showed greater accumulation of proline in the shoots than the -AMF plants, comparing them within the same water regime. The highest accumulation was observed in plants under +SD, which were about 54% higher than plants without AMF in the same water regime. However, *H. serratifolius* plants under +MD and +WW showed an approximate 60% increase of proline in the shoot compared to plants under -MD and -WW (Figure 2A). The accumulation of proline in the root was mainly influenced by the water regime, where plants in the SD had the highest accumulations, about 80% higher than plants in the MD. However, in the ideal water regime, +WW plants showed 70% higher accumulation than -WW plants (Figure 2B).

Plants within the same water regime showed no statistical differences for the accumulation of ammonium (*p* ≤ 0.05); only the -WW treatment provided an increase of about 45% for the plants in the +MD and +WW treatments (Figure 3A). Regarding nitrate accumulation, it was higher in plants under +WW (about 60% higher than in plants without AMF), but there were no differences in the other treatments (Figure 3B).

### 2.3. Phosphorus

Regarding phosphorus (P), well-hydrated plants in the presence of AM fungi had the highest concentration of phosphorus in the root, while under water deficit conditions there was no statistical difference (*p* ≤ 0.05) (Figure 4).

### 2.4. Carbohdrate Metabolism

It was observed that under the +MD regime, the plants of *H. serratifolius* showed a higher accumulation of total soluble sugar at around 45% compared to the plants under the –MD regime. In the WW and SD regime, the +AMF plants also showed slightly higher accumulation than -AMF plants (Figure 5A). Plants submitted to +WW showed a 64% higher accumulation of non-reducing sugar than plants in -WW (Figure 5B). In SD, -AMF plants showed an increase of 29% in relation to +AMF plants. In MD, AMF had no significant effect (*p* ≤ 0.05). Plants in SD, regardless of the presence of AMF, showed the highest values of reducing sugar accumulation, followed by plants in -MD, which showed accumulation 18% higher than +AMF plants in the same water regime. In WW, the presence of AMF did not have a significant effect (*p* ≤ 0.05) (Figure 5C).

## 3. Discussion

The presence of AM fungi encouraged the accumulation of osmolytes derived from nitrogen and carbohydrate metabolism, providing greater growth and absorption of phosphorus and nitrogen compounds in *H*. *serratifolius* plants under different water regimes. With regard to growth, Zou et al. [24] and Zhang et al. [25] observed an increase in shoot dry mass and stem growth in seedlings of *Poncirus trifoliata* and *Zenia insignis,* respectively, when in the presence of AMF, both in ideal water conditions and in a water deficit.

The influence of AMF on leaf development is variable; Zhang et al. [26] observed that, in *Cyclobalanopsis glauca*, in the presence of AMF, there was an increase in leaf area only under water deficit conditions, while Zou et al. [24] observed in *Poncirus trifoliata* there were benefits of AMF on the number of leaves in well-hydrated plants and under a water deficit.

However, Vieira et al. [27] did not observe any influence of water deficit on leaf development in *H. serratifolius*. In our study, the total leaf area and the number of leaves were not influenced by the water deficit due to the short period of time of exposure to stress. Plants subjected to water stress have different responses that include increased root/shoot ratio, reduced growth, altered leaf anatomy, and reduced leaf size, as well as a reduced total leaf area to limit water loss and ensure photosynthesis [28].

Regarding root volume, Zou et al. [24] reported an increase in root volume when comparing *P. trifoliata* seedlings infected with AM fungi to seedlings without AM fungi, both in ideal water regimes and water deficits, in order to verify that the root volume was higher in conditions of ideal hydration than in water deficit. This scenario was also observed by Huang et al. [29] in *Malus prunifolia* (Willd.) Borkh. and corroborated in this study.

One of the benefits of AMF is the colonization of plant roots, increasing the area explored by the root. Urgiles et al. [30] reported that colonization of tree roots from forest species can reach 70%. For the genus *Handroanthus*, the %C ranged from 13% to 47% [31]. Frosi et al. [32] observed in *Poincianella pyramidalis* under different water regimes a %C ranging from 31.8% to 29.9%, with no statistical differences, whereas in our study, %C in *H. serratifolious* ranged from 17.3% to 57.7%, with the highest %C in +WW condition. A decrease in %C by AMF due to the increase in water deficit was also reported by Zhang et al. [25] and Begum et al. [33] in seedlings of *Zenia insignis* and *Zea mays*, respectively. The decrease can be attributed to the lower availability of moisture in the soil, impairing both the development of AMF and vegetables [33]; in the present work, a decrease in root volume was observed as well as an increase in water limitation, even more accentuated in plants in the –SD condition.

The benefits to plant growth are related to several physiological, biochemical, and nutritional factors, such as an increase in net photosynthetic rate, greater PSII efficiency, greater accumulation of osmolytes such as proline, soluble sugars, and total amino acids [34]. It is also possible to observe a greater absorption of P and nitrogen compounds through greater activity of some enzymes [33]. In some cases, it may be due to morphological changes with increasing root volume and dry mass [35]. Zou et al. [24] and Zhang et al. [25] observed in seedlings of *Poncirus trifoliata* (L.) Raf. and *Zenia insignis* Chun, respectively, increases in shoot dry mass and stem growth when cultivated in the presence of AM fungi, both in ideal water regimes and in water deficits.

The total protein results of the present study were similar to those observed by Oliveira et al. [36] and Baslam et al. [37], in which they did not find a significant increase in total protein content in seedlings of *Myracrodruon urundeuva* M. Allemão and *Lactuca sativa* L., respectively, which were inoculated with AM fungi under ideal hydration conditions. However, the highest accumulations of total proteins were observed in plants under water stress, justified by the increase in the production of enzymes of the antioxidant system to combat reactive oxygen species.

The maximum possible content of soluble proteins in plants not inoculated with AM fungi under water stress is also related to the content of total amino acids, as the plants with the lowest protein values were those that presented the highest values of amino acids. The opposite was also observed, in which the plants with the highest protein values had the lowest amino acid values. The increase in total free amino acid content is a strategy plants use to tolerate water stress, as the accumulation reduces the osmotic potential by increasing concentrations and, thus, increasing cellular water retention during water stress [17]. The main amino acids related to osmotic control are proline, mannitol, trehalose (Thre), D-inositol, sorbitol, betaine β-alanine, polyamines (Pas), and dimethyl sulfonium propionate [38].

The accumulation of proline in plants under water deficit is known, but the presence of AM fungi changes its dynamics. Higher levels of proline were observed in seedlings of *Macadamia tetraphylla* L.A.S.Johnson [39], *Ephedra foliata* Boiss. ex C.A.Mey. [40] and in the present study in *H. serratifolius*, but a decrease in proline content was observed in *P. trifoliata* [1,24]. Proline biosynthesis has been associated with the glutamate or ornithine pathways [41], which are both amino acids that that were significantly increased in plants with AM fungi.

Proline acts as an osmotolerant agent, in addition to being an energy source [42] and generates, during its catabolism, the energy equivalent to about 30 ATP molecules [43] and nitrogen molecules [40,44], which is a fundamental process for stress recovery. Another important role of proline is in maintaining tissue water content under water deficit conditions [41], protecting plant proteins and membranes from damage caused by excess reactive oxygen species [45]. Thus, the accumulation of proline in *H. serratifolius* plants is of fundamental importance both to tolerate water deficits, and to aid in recovery after stress, providing greater gain in dry mass, as observed in our study.

For most higher plants, ammonium (NH_4_^+^) and nitrate (NO_3_^−^) are the two main ways to absorb inorganic N from the soil [46]. They are compounds that can play different roles when plants are under water deficit [29,33]. The greater accumulation of nitrate is related to the greater activity of nitrate reductase and nitrite reductase in AMF plants [47]. During the assimilation process, NO_3_^−^ is converted into NH_4_^+^ by nitrate reductase (NR) and nitrite reductase (NiR). Subsequently, NH_4_^+^ is assimilated to glutamine and glutamate via glutamine synthetase (GS) and glutamate synthase (GOGAT) [48,49]. NR catalyzes the rate-limiting step of nitrogen assimilation, and increased activity has been reported to directly influence the synthesis of key stress-protective amino acids such as proline, thus leading to the regulation of important physiological processes under stress [50].

The results in P were similar to those observed by Pavithra and Yapa [51]. In general, the presence of AM fungi promotes greater absorption of this nutrient, which is widely documented in the literature with several plant species such as *Z. insignis* [24], *C. glauca* [26], *Ipomoea batatas* (L.) Lam. [52]. and *Hevea brasiliensis* (Willd. ex A.Juss.) Mull.Arg [53]. However, it is worth mentioning that in all these works, the evaluation of phosphorus content was carried out in the leaves, while in the present work it was carried out in the root. In this study, under conditions of good hydration, the phosphorus content in the roots of mycorrhizal +AMF plants was about 200% higher than in -AMF plants.

Thus, with the imposition of the water deficit and, consequently, the decrease in soil moisture, there was a decrease in the absorption of P by the roots, even in plants with mycorrhiza, since this nutrient is immobile in the soil and limits the normal growth of plants [42]. Often the higher phosphorus uptake in plants with AM fungi is related to a greater surface area for uptake provided by fungal hyphae [54], and to the induction of the formation of genes and plant phosphate transport proteins, expressed in the hyphae outside the root [55,56,57], in addition to the increase in alkaline and acid phosphatase enzyme activity [40].

Several studies have shown greater accumulation of soluble sugar in plants with mycorrhiza under stress conditions, mainly moderate stress [25,58,59,60]. Studies such as Zarik et al. [61] with *Cupressus atlantica* Gaussen also observed the highest accumulation of soluble sugar in a condition of 25% FC, as was the case in our work.

Carbohydrate accumulations change according to developmental stage, environmental factors, and species. Yooyongwech et al. [52] did not observe an increase in reducing and non-reducing sugars in the leaves of *I. batatas* in the presence of AM fungi compared to plants without AM fungi. Tisarum et al. [53], in *H. brasiliensis*, observed results similar to those found in this study, since under conditions of good hydration, they observed an increase in non-reducing sugar in plants with AM fungi, while under stress conditions, plants without AM fungi showed higher sucrose content.

Carbohydrates participate in energy metabolism by regulating plant growth and development, acting as important molecules in the regulation of stress responses and tolerance mechanisms, mainly in the form of osmolytes [34,62]. Water deficit conditions almost always result in changes in the levels of reducing sugars, such as glucose, fructans, and non-reducing sugars such as sucrose and raffinose [63]. Additionally, water deficit conditions can increase the enzymatic activity of sucrose phosphate synthase, neutral invertase, and the net activity of sucrose-metabolized enzymes in leaves, and decrease enzymatic activity of acid invertase and sucrose synthase in leaves [1].

Sucrose is the main sugar translocated via phloem, from the source, mainly leaves, to the drain; in plants with AM fungi, the root becomes a strong drain. In the root, sucrose is converted to glucose and fructose, where glucose can then be routed to the AM fungi and both molecules can function as osmolytes [34]. Sucrose acted more strongly as a source of energy for the growth of plants with AM fungi under conditions of good hydration. Free proline and soluble carbohydrates are the main osmolytes that act against water stress in higher plants, such as *Olea europaea* L. [64], *P. trifoliata* [65], and *M. tetraphylla* [38].

Water deficit induces several physiological, biochemical, and molecular changes in plants that lead to increased plant tolerance [28]. Normally, it increases the root-shoot ratio and reduces water consumption by reducing leaf area [66]. However, in the present study, biochemical changes were mainly ones observed to increase tolerance of water deficits, such as increased accumulation of proline, total amino acids, sugars, and nitrate, and not ones that strongly influence morphological changes in *H. serratifolius* plants.

These biochemical changes contribute to reducing the osmotic potential and, therefore, the leaf water potential in plants exposed to drought [35]. The lower value of the water potential allows the plant, with mycorrhiza, to sustain the high level of hydration and turgor of the organs, which maintain the general physiological activities of the cells and is especially linked to the photosynthetic apparatus [67].

## 4. Materials and Methods

### 4.1. Location and Experiment Design

The AM fungi used in this study were obtained from the rhizospheric soil of an area of savanna located in the region of Alter do Chão, in Santarém city, west of Pará state, Brazil (2°28′1″ S and 54°49′41″ W). The soil in the area is predominantly composed of oxisol, which is well-drained, acidic, deficient in phosphorus and has low natural fertility, with intense weathering and high iron and aluminum oxide content [68]. The infective inoculum was produced from rhizospheric soil samples as described by Santos et al. [69].

The soil used as substrate was disinfected in an autoclave as recommended by Santos et al. [70] and evaluated for its chemical and granulometric properties [71] (Table 2).

The experiment was carried out according to a bifactorial organization in a completely randomized design. The first factor, which represented the inoculation status of plants, had two levels: (1) + AM = inoculated; and (2) -AM = non-inoculated. The second factor was the water regime with three levels, according to Mo et al. [60], with changes defined from previous studies: (1) 10% field capacity (FC) simulating severe water deficit (SW); (2) 25% (FC) simulating moderate water deficit (MD); and (3) 62% (FC) simulating a well-watered (WW) condition. The combination of the gradients of the two factors resulted in 6 experimental variants (2 × 3). The experiment was carried out in eight replicates, resulting in 48 pots (6 variants × 8 replicates).

The cultivation was carried out at the Laboratory of Plant Physiology and Plant Growth, at the Federal University of Western Pará. The seeds of *H. serratifolius* were sown in pots with a capacity of 0.7 L, in a controlled environment with daily irrigation to maintain the initial field capacity of 62%. Each repetition of the group of mycorrhizal plants received 25 g of infective inoculum, which had a density of 30.89 spores per gram, with the Glomeraceae family being the most frequent. 30 days after sowing, the implementation of the water deficit began; irrigation was reduced until reaching a field capacity of 10% and 25%. After 60 days of sowing, all seedlings were submitted to different water regimes of 10% (SD), 25% (MD), and 62% (WW), a situation that remained until the 90th day.

### 4.2. Observations and Analysis

The growth analysis of *H. serratifolius* seedlings took place on the 90th day, where shoot height and root volume were quantified as described by Santos et al. [69]. The leaf area (LAI) was obtained by the formula LA = L × W × F, where, (L) is the leaf length and (W) leaf width and (F) is the correction factor for the leaves of *H. serratifolius* (0.6206). The correction factor “F” was determined by simple regression analysis between the area of a sample of leaves and the product of its dimensions. In this case, the line fitted to the data set had an equation of the type Y = bx, where “b” corresponds to the factor “F” (Figure 6). The actual leaf area sampled was determined by digitizing the respective images and later calculating the area with ImageJ software [72]. The factor was then tested and validated using regression analysis, examining the relationship between the estimated leaf area and the real leaf area in a new leaf sample.

A fresh root sample of approximately 1 g of each treatment was taken to evaluate colonization by AM fungi, and the quantification of the percentage of colonization was performed according to Phillips and Hayman, [73] and Giovannetti and Mosse [74]. Afterwards, the plants were placed in an oven with forced air circulation at a temperature of 60 °C until at a constant weight. Later, the samples of shoot and root dry mass were weighed separately on an analytical balance with a precision of 0.01 g, and samples were then packed for further analysis.

A part of the dehydrated samples was taken to quantify the contents of total soluble sugars by the Antrona method [75], reducing sugars by the DNS method [76], and the non-reducing sugars [77], total proteins [78], total soluble amino acids [79] and proline [80]. The values of free ammonium [81], free nitrate [82], and phosphorus content were measured from the extraction by Mehlich-1 after digestion in a muffle furnace at 500 °C, and the determination was carried out by an acidic ammonium molybdate solution [83]. To calculate the accumulation, the dry mass was multiplied by the content.

Statistical analyses were performed using the SISVAR program version 5.8.92 [84], where the variables were subjected to normality verification by the Shapiro-Wilk test (*p* > 0.05) and the means were compared by the Tukey test at the 5% level of significance.

## 5. Conclusions

Due to our study and the current literature, we propose that osmoregulation in *H. serratifolius* plants with AM fungi in moderate water deficits is performed primarily by soluble sugars and total soluble amino acids, while in severe water deficits, it is controlled by proline and reducing sugars. AM fungi promoted greater absorption of phosphorus and nitrate only under a well-watered regime. Furthermore, AM fungi provided an increase in dry mass in both water deficit and well-watered regimes, but plant height was more strongly affected by AM fungi only under the well-watered regime. Our findings showed that the mycorrhizal colonization was directly related to the water regime, where well-watered plants had a higher percentage of colonization, and plants with severe water deficits decreased the colonization percentage.

## Figures and Tables

**Figure 1 plants-11-02731-f001:**
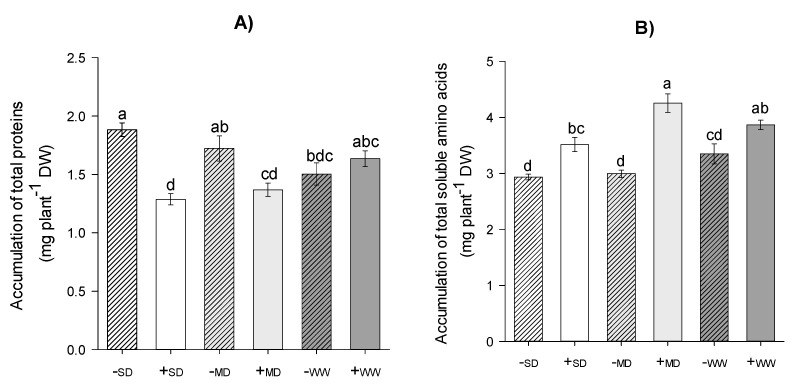
Accumulation of total proteins (**A**) and total soluble amino acids (**B**) of *H. serratifolius* plants subjected to different water regimes in the presence and absence of arbuscular mycorrhizal fungi. -SD: severe water deficit without AMF; +SD: severe water deficit with AMF; -MD: moderate water deficit without AMF; +MD: moderate water deficit with AMF; -WW: well-watered without AMF; +WW: well-watered with AMF. The means of variables with the same letter are not statistically different by Tukey’s test ≤ 5%.

**Figure 2 plants-11-02731-f002:**
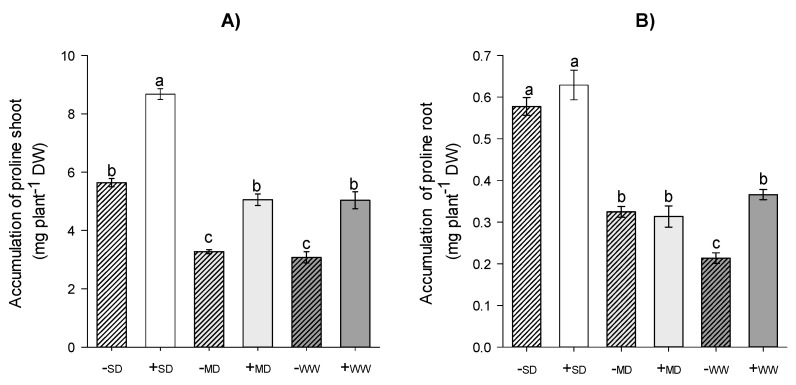
Variation in shoot (**A**) and root (**B**) proline accumulation of *H. serratifolius* plants submitted to different water regimes in the presence and absence of arbuscular mycorrhizal fungi. -SD: severe water deficit without AMF; +SD: severe water deficit with AMF; -MD: moderate water deficit without AMF; +MD: moderate water deficit with AMF; -WW: well-watered without AMF; +WW: well-watered with AMF. The means of variables with the same letter are not statistically different by Tukey’s test ≤ 5%.

**Figure 3 plants-11-02731-f003:**
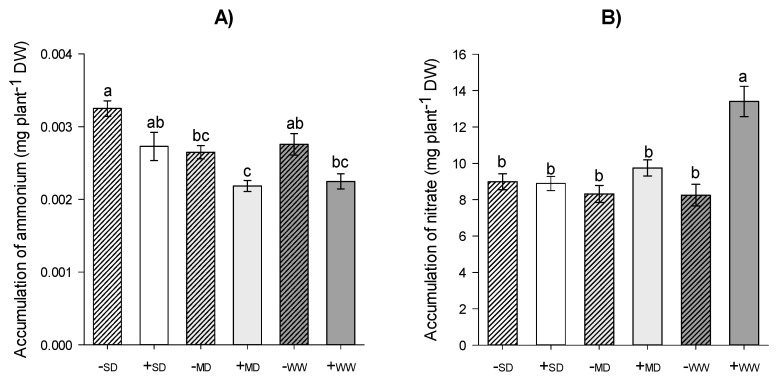
Accumulation of ammonium (**A**) and nitrate (**B**) of *H. serratifolius* plants submitted to different water regimes in the presence and absence of arbuscular mycorrhizal fungi. -SD: severe water deficit without AMF; +SD: severe water deficit with AMF; -MD: moderate water deficit without AMF; +MD: moderate water deficit with AMF; -WW: well-watered without AMF; +WW: well-watered with AMF. The means of variables with the same letter are not statistically different by Tukey’s test ≤ 5%.

**Figure 4 plants-11-02731-f004:**
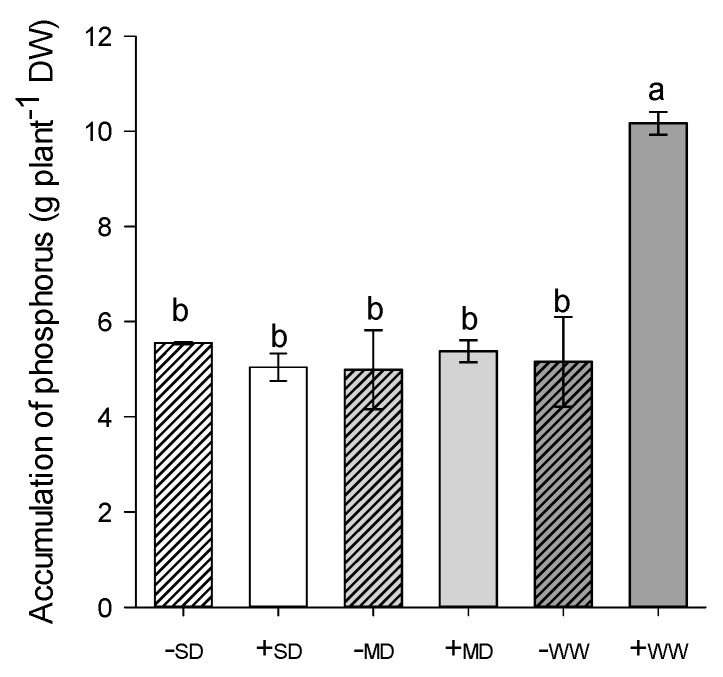
Accumulation of phosphorus of *H. serratifolius* plants subjected to different water regimes in the presence and absence of arbuscular mycorrhizal fungi. -SD: severe water deficit without AMF; +SD: severe water deficit with AMF; -MD: moderate water deficit without AMF; +MD: moderate water deficit with AMF; -WW: well-watered without AMF; +WW: well-watered with AMF. The means of variables with the same letter are not statistically different by Tukey’s test ≤ 5%.

**Figure 5 plants-11-02731-f005:**
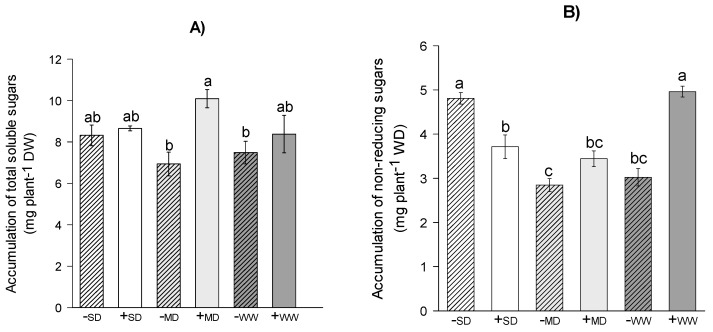
Total soluble sugar (**A**), non-reducing (**B**) and reducing (**C**) sugar of *H. serratifolius* plants submitted to different water regimes in the presence and absence of arbuscular mycorrhizal fungi. -SD: severe water deficit without AMF; +SD: severe water deficit with AMF; -MD: moderate water deficit without AMF; +MD: moderate water deficit with AMF; -WW: well-watered without AMF; +WW: well-watered with AMF. The means of variables with the same letter are not statistically different by Tukey’s test ≤ 5%.

**Figure 6 plants-11-02731-f006:**
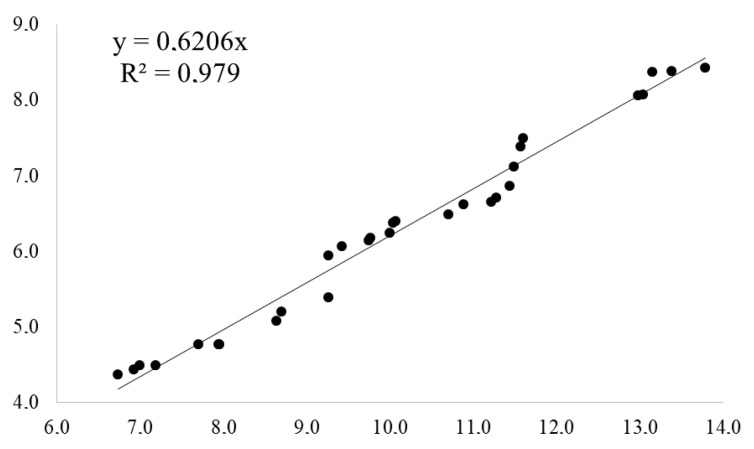
Simple regression equation between the area of the *H. serratifolius* leaves and the product of their dimensions, resulting in a correction factor of 0.6206.

**Table 1 plants-11-02731-t001:** Growth and mycorrhizal colonization of *H. serratifolius* plants subjected to different water regimes in the presence and absence of arbuscular mycorrhizal fungi.

Water Regime	Inoculation	Shoot Dry Mass (g)	Shoot Height (cm)	LA (cm^2^)	TLA (cm^2^)	RV (cm^2^)	%C
SD	-AMF	82.3 b	10.6 ab	5.41 a	37.7 a	0.27 c	-
	+AMF	86.7 ab	9.99 ab	5.23 a	38.0 a	0.37 bc	17.3 c
MD	-AMF	81.1 b	10.5 ab	5.50 a	40.0 a	0.32 bc	-
	+AMF	93.6 a	10.8 a	6.83 a	47.8 a	0.33 bc	29.8 b
WW	-AMF	79.9 b	9.53 b	5.36 a	38.9 a	0.53 a	-
	+AMF	94.9 a	10.7 a	6.88 a	44.7 a	0.44 ab	57.7 a

SD: severe water deficit; MD: moderate water deficit; WW: well-watered; -AMF: absence of AMF; +AMF: presence of AMF; LA: leaf area; TLA: total leaf area; RV: root volume; %C: colonization percentage; - absence of mycorrhizal colonization. The means of variables with the same letter are not statistically different by Tukey’s test ≤ 5%.

**Table 2 plants-11-02731-t002:** Chemical properties of soil used as substrate.

pH	Al	Ca	Mg	H + Al	H	P	K	Fe	Mg	MO	CO	Areia	Argila	Silte
	cmolc/dm^3^					mg/dm^3^				dag/dm^3^		%		
5	2.42	1.03	0.53	5.3	2.9	22	102	65.1	6.3	5.47	3.18	42.9	41.5	15.6

pH: hydrogen potential; Al: aluminum; Ca: calcium; Mg: magnesium; H + Al: calcium acetate; H: hydrogen; P: phosphorus; K: potassium; Fe: iron; Mg: magnesium; MO: organic matter; CO: organic carbon.

## Data Availability

Not applicable.

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
