# Peer review of "Arbuscular Mycorrhizal Fungi Promote Physiological and Biochemical Advantages in *Handroanthus serratifolius* Seedlings Submitted to Different Water Deficits"

_plants, 2022, doi:10.3390/plants11202731_

Round 1

Reviewer 1 Report

This manuscript descrives the impact of MA fungi on the initial growth of seedlings of H. serratifolius seedlings under several water regimes.

The introduction is complete clear and states clearly the goal of the research.

The material and methods and clearly descrived.

The results and discussion section is clearly preseted and the discussion goes to plenty of details along the results.

The conclusion are based on the results.

Nothing particular to say to improve it.

Reviewer 2 Report

The manuscript with the title “Arbuscular Mycorrhizal Fungi Promote Physiological and Bio-chemical Advantages in Handroanthus serratifolius Seedlings Submitted to Different Water Deficits”, investigated the potential benefits of arbuscular mycorrhiza inoculation for establishment of Handroanthus serratifolius (famlily Bignoniaceae), in the prospects of reforestation. This plant grows in the cerrado vegetation of South America, and is one of the strongest and largest tropical forest trees. Therefore, the research is highly relevant and has immediate applications.

Introduction

Line 46 - Arbuscular Mycorrhizal = (AM), please see lines below (50, 63, 67, 68, 70 …) where you used MA instead of AM. Replace MA with AM.

At the end of the introduction section the aim and objectives were not clearly defined. Also consider that Conclusions have to mirror the objectives, where objectives are defined as the steps considered to reaching the aim, and the conclusions are written to answer to each of them based on results.

Material and Method

Please divide Material and Method section in at least two chapters: 1) Location and experiment Design, 2) Observations and Analysis.
Please also add a section of “Biologic material” if you consider necessary, or at least give some details about the provenance of the seeds used in the experiment.

Line 312 “The experiment was carried out in a completely randomized design, where the plants were divided into two groups …” I suggest to rephrase such as: “The experiment was carried out according to a bifactorial organization in completely randomized design. First factor represented by inoculation status of plants had two levels: 1) + AM = inoculated, 2) -AM = non-inoculated. The second factor was the water regime with three levels: 1) 10% field capacity (FC) simulating severe water deficit (SW), 2)… 3)… From the combination of the gradients of the two factors resulted 6 experimental variants (2×3). The experiment was carried out in eight replicates, resulting 48 pots (6 variants x 8 replicates).”
Could it be possible to add a visual representation of the experiment layout? Such as a graph or flow-chart (in a figure). It would be more easily for the readers to understand the experiment.

Results

Please divide the Results section into chapters, in order to group information into a way that is easy for readers to search into text for what it interests them the most.

From table it seems like the greater the water deficit, the lower the colonization level. This is interesting but somehow surprising.
One might expect that dependency of the plants increases to AM under water deficit, while under well-watered conditions the Carbon cost for the plant might be less justified, thus triggering some suppression of colonization, maintaining it to lower levels by the host.  Due to this result I want to ask if the authors determined the AM colonization morphotype (Arum, Paris, Intermediate or combined) present in the roots, that might also play an important role in the outcome. Are there any studies on species from same botanic family or tropical trees from Brazil? And if yes, what kind of mycorrhizal status and colonization was determined for those. Are these colonization levels low or high compared to those species?

For Discussion

When discussing the results, it is very important to choose literature in a sense that is most relevant for understanding or supporting the results being presented. Therefore, it is good that authors tried to cite literature both that lead to similar results or contrasting to these. However, it is very important to be careful in regards with species chosen, that have to be considered: life form (herbs vs trees have very different life strategy and metabolic pathways), phylogenetic closeness of taxa (compare results with plants from same family such as Catalpa sp. for example). Therefore, comparing the results obtained for a tropical tree species (a phanerophyte) with lettuce and soy (therophytes) I personally consider less relevant, and unable to highlight/support the most important aspects conveyed. Please also find sources for colonization level of related tress or tropical trees, to know if these colonization values are low or high. The AM literature is wide but please if you find a few highly relevant papers that can provide most relevant results comparable to what was done here, please cite them.

Best regards.

Reviewer 3 Report

I have read carefully the manuscript “Arbuscular Mycorrhizal Fungi Promote Physiological and Biochemical Advantages in Handroanthus Serratifolius Seedlings Submitted to Different Water Deficits”. The topic of the study is interesting and under the scope of the Journal but doesn't have any novel point. No doubt, the authors did good work but the study is not in-depth, as the mechanism of AMF in drought stress is missing. Authors mainly focus on osmolytes measurements, not any linked physiological and biochemical analysis. A lot of grammatical and typo mistakes throughout the manuscript, and many sentences have no structure. The introduction does not support the study background, and the discussion part is also weak.

Here are some specific suggestions and concerns

-Abstract is not well written. No concluding remarks at the end of the Abstract. It should start with an attractive sentence and an overall summary of the background, methodology, results, and concluding remarks.

-Introduction is not explaining well the background of the study. As authors measure the osmolytes but do not explain the relationship of AMF with osmolytes and the role of osmolytes in drought tolerance.

-Authors used the term “MA fungi” throughout the manuscript. There is no meaning for MA fungi, so I suggest changing it to AM fungi.  

-Author should give justification why he chooses 10%, 25%, and 62% field capacity for water stress.

- It's better to write results and discussion separately.

-Why do you measure only root volume and not measure Root length, Root dia., etc?

-Why do you measure only proline from roots and not measure the protein, sugar, and nutrients?

-Conclusion is confusing, and the explanation of the study findings and concluding remarks should be clear.

-What is total soluble amino acid? Which you stated in figure 1B.

-In Figure 1, Change the X-axis label “Accumulation total proteins” to “Accumulation of total proteins” and check other labels also.

Round 2

Reviewer 2 Report

My comments were satisfactory addressed by the authors.

Author Response

Thanks fot the suggestions. 

Reviewer 3 Report

The authors have tried to improve the manuscript. Still, improvement is required.

The authors didn’t respond to my previous file comment “In Figure 1, Change the X-axis label “Accumulation total proteins” to “Accumulation of total proteins” and check other labels also”. Check carefully all figures and correct the mistakes.

Line 24: MA fungi??? Again, the same mistake.

Line 336: defined from previous studies, Cite the reference.

Line 392: Change the sentence “We also found that mycorrhizal colonization is directly related to the water regime, where well-watered plants have a higher percentage of colonization and plant with severe water deficit the lower %C” to “Our findings showed that the mycorrhizal colonization is directly related to the water regime, where well-watered plants have a higher percentage of colonization whereas plant with severe water deficit decreases the colonization percentage”.

Author Response

Por favor, verifique o anexo.
